# Revisiting suicide rate during wartime: Evidence from the Sri Lankan civil war

Takeshi Aida ORCID *

Institute of Developing Economies, Japan External Trade Organization (IDE-JETRO), Chiba, Japan

* aidatakeshi@gmail.com

**Data Availability Statement:** All relevant data are within the manuscript and its Supporting Information files.

**Funding:** TA received JSPS KAKENHI Grant 18K12786 from Japan Society for the Promotion of Science (https://www.jsps.go.jp/english/index.

## Abstract

After the seminal work of Durkheim (1897), many subsequent studies have revealed a decline in suicide rates during wartime. However, their main focus was inter-state wars and whether the same argument holds for civil conflicts within a country is an important unresolved issue in the modern world. Moreover, the findings of the previous studies are not conclusive due to unobserved confounding factors. This study investigated the relationship between civil war and suicide rate through a more rigorous statistical approach using the Sri Lankan civil war as a case study. For this purpose, we employed a linear regression model with district and year fixed effects to estimate a difference-in-difference in the suicide rate between the peacetime and wartime periods as well as the contested and non-contested districts. The results indicate that the suicide rate in the contested districts in the wartime was significantly lower than the baseline by 11.8–14.4 points (95% CI 6.46–17.22 and 7.21–21.54, respectively), which corresponds to a 43–52% decline. The robustness of the possible confounding factors was analyzed and not noted to have so much effect as to alter the interpretation of the results. This finding supports the Durkheimian theory, which places importance on social integration as a determinant of suicide, even for civil conflicts.

## Introduction

At the end of the 19th century, Durkheim proposed that suicide is a social phenomenon [1]. He argued that suicide rates are negatively related to the degree of social integration and regulation, whereas over-integration or over-regulation also induces other types of suicide. Wartime is considered to be the period when the degree of social integration increases, leading to lower suicide rates than those in peacetime. Specifically, he stated, "So there can only be one explanation for these facts, which is that great upheavals in society, like great popular wars, sharpen collective feelings, stimulate the party spirit and the national one and, by concentrating activities towards a single end, achieve, at least for a time, a greater integration of society" [1]. Many subsequent studies have also confirmed the decrease in suicide rates during wartime [2–7]. However, the theory needs a more detailed examination in terms of several conceptual and technical perspectives.

One of the most important issues is that the nature of war has undergone a sea change from Durkheim's era. While Durkheim assumed inter-state wars in his argument, civil wars have

html). The funders had no role in study design, data collection and analysis, decision to publish, or preparation of the manuscript.

**Competing interests:** The authors have declared that no competing interests exist.

become more common than inter-state wars in the modern world [8]. Meanwhile, several studies have demonstrated that civil wars harden ethnic identities and contribute to a growing polarization of society, converting local disputes into a national schism [9–11]. Thus, it is possible that civil wars increase the level of integration though their bodies are smaller units (e.g., ethnic groups) than the theory originally assumed.

The consequences of civil conflicts themselves are not a new research theme. Several previous studies have shown that exposure to civil conflicts has a potentially long-lasting negative effect on people's health [12–14]. However, their focus is on the effects on physique and disability, and the effect on suicide rate remains largely unexplored. Thus, whether civil wars also lead to a decrease in suicide rates is an important issue to be investigated in the field of public health.

Another important issue is that the previous studies rely mainly on time-series variations to detect the decrease in suicide rates during wartime. However, such an approach can be prone to omission of confounding factors. For example [15], showed that the decline in suicide rate in the US during World War II was spurious for failing to take into account the decline of the unemployment rate during wartime. In addition [16], showed that the Scottish suicide rate increased during World War II when the overall declining trend is taken into account. These studies suggest that more rigorous statistical analysis is essential to test whether wars per se lead to the lower suicide rate.

This study aims to explore the effect of the Sri Lankan civil war on the suicide rate. Sri Lanka experienced a civil war between the government (GOSL) and the Liberation Tigers of Tamil Eelam (LTTE), a militant organization that aimed to create an independent state of Tamil people, from 1983 to 2009 [17]. Sri Lanka is also known for its high suicide rate: it was among the highest in the world though it has been decreasing since 1996 [18, 19]. Therefore, suicide in Sri Lanka is a crucial issue to be explored with regard to academic and policy perspectives, and many studies have been conducted concerning this issue [20].

The relationship between suicide and the civil war in Sri Lanka has not been fully investigated. Furthermore, even among a few studies addressing this issue, there has been no solid consensus on whether the Sri Lankan civil war led to the lower suicide rate. On one hand, for example, one study demonstrated the decrease in suicide rate during the war in Jaffna district, where the armed conflicts were the severest in the country [21]. On the other hand, another study attributed the decrease to banning pesticide rather than the civil war as pesticide poisoning has been the most preferred method for suicide in the country [18]. However, as is often the case with the studies on this issue, both of these studies rely only on the time-series variation in suicide rate, and thus the effect might be confounded with many unobserved factors. Furthermore, the intensity of conflicts was not uniform within the country: the majority of the fighting took place in the northern and the eastern districts, where LTTE exerted territorial control. Therefore, a cross-sectional comparison to uncover the relationship between civil war and suicide will also be informative.

This study employed more rigorous statistical analysis than previous studies—to incorporate the above-mentioned issues—by estimating the difference in the suicide rate between the contested and the non-contested areas in the country, as well as the difference between peacetime and wartime. This approach enabled us to control for unobserved heterogeneities on the assumption that they are either time-invariant or parallel between contested and non-contested areas. Moreover, the analysis carefully considered several possibilities that might alter the interpretation of the results. Therefore, the current study is a more definitive case study on the classic issue of whether a war leads to a lower suicide rate in the sense that it provides evidence robust to possible confounding factors.

## Materials and methods

### Data

This study used several data sources on the suicide rate at the district level. The suicide rates in the pre-war period were obtained from the previous studies [22, 23], which were originally obtained from the Registrar General's Department and have been used by several studies on this issue [18, 24]. Their data include district-level (i.e., the second-level administrative divisions in Sri Lanka) suicide rate per 100,000 population, covering the years 1955, 60, 65, and 70–80. The suicide rate by sex is also available for the years 1955, 65, 72–75, and 80. The suicide rates during and after the civil war were obtained from *Statistics on Vital Events 2000–2010* and *Vital Statistics Report*, both of which were issued by the Registrar General's Department, Ministry of Public Administration and Home Affairs Sri Lanka in 2011 and 2018. The former includes the number of incidents of suicide at district level from 2000 to 2006 and the estimated population size so that we can calculate the suicide rates. The latter includes the same information for the years 2007, 2009, 2010, and 2013.

Under the Births and Deaths Registration Act, the registration of births and deaths is compulsory in Sri Lanka, and refusal to answer or providing false information can be subject to imprisonment or a fine [25]. The Registrar General's Department has a registrar in each registration division, which is further divided from each administrative district. Although suicide statistics of developing countries are often subject to skepticism, these statistics of Sri Lanka are thus thought to be about as reliable as those from many developed countries [22].

However, there remain several reasons to be skeptical about the reliability of the data, especially during wartime. For example, suicide is stigmatized in Sri Lanka, as in many other countries: the suicide of a family member exposes the subterranean family problems and might diminish the prospects of marrying off their children [26]. Thus, one might be concerned that such stigma may provide bereaved families with an incentive to misreport suicides as fatalities via the civil war. However, according to the Criminal Procedure Code, unnatural deaths, including suicides, are subject to a coroner's inspection before the cause of death is concluded [27]. Therefore, it is unlikely that fatalities via the war are confounded with the ones from suicide although the possibility cannot be ruled out in a strict sense. Although we examined the issue of data reliability during wartime in the sensitivity analysis, the results should still be interpreted with caution.

The data of covariates were obtained from *Statistical Abstract*, which is an official statistical report issued by Department of Census and Statistics (almost) each year. The numbers of victims in each armed conflict were obtained from the Uppsala Conflict Data Program (UCDP), which is a well-known dataset for civil war [28].

One problem of constructing the panel dataset at the district level was that the administrative division of districts changed from the pre-war and wartime periods: Gampaha was carved out of the northern part of Colombo in September 1978; Kilinochchi was carved out of the southern part of Jaffna in February 1984; Mullaitivu was carved out of the northern part of Vavuniya together with parts of the then Jaffna, Mannar, and Trincomalee in September 1978. For these newly divided districts, the data were collapsed with that of the original districts to construct a panel dataset.

Fig 1 illustrates the districts that LTTE claimed as Tamil Eelam—an independent state of Tamil—as of June 2006 [29]. These territories varied in the course of the civil war but were almost stable during the sample period from 2000 to 2006 due to the ceasefire agreement in February 2002. Hereafter, we label these districts as "contested districts" regardless of whether it is in the pre-war or the wartime period.

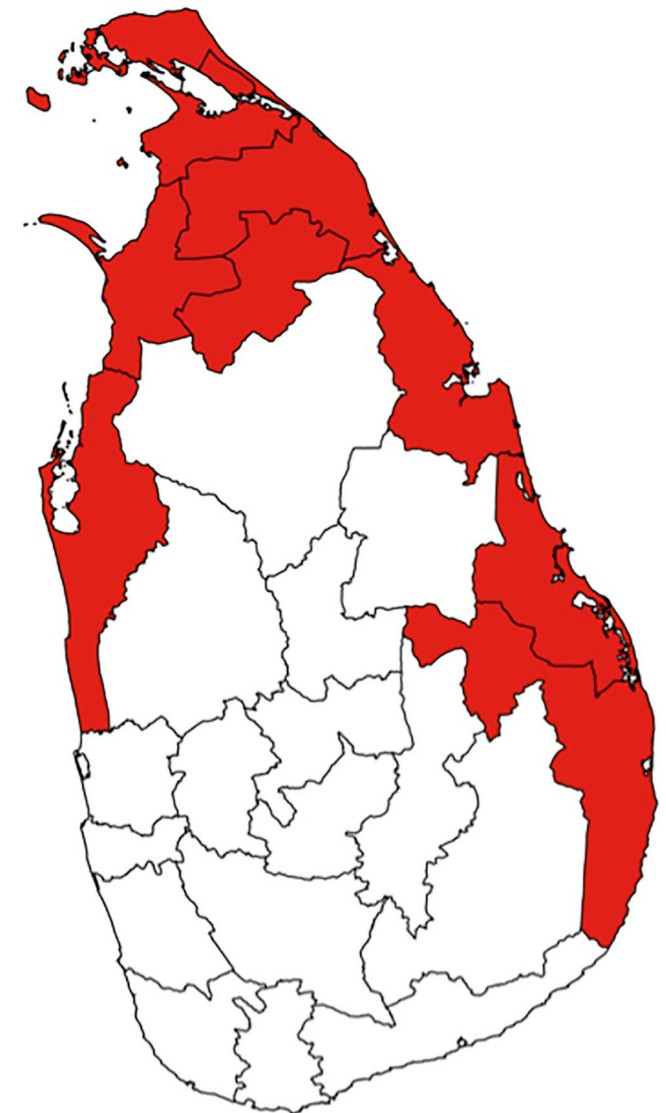

**Fig 1. The districts where LTTE claimed as Tamil Eelam.** The figure illustrates the districts which LTTE claimed as Tamil Eelam as of June 2006. (Source: [29]) These territories were almost stable during the sample period of 2000–2006.

A substantial difference exists in the intensity of the conflicts between the contested and the non-contested districts. According to the best estimate of the UCDP dataset, there were 39 incidents in the non-contested area from 2000 to 2009, whereas this number was 68 in the contested area. The difference in the number of victims is quite substantial. The average number of deaths per incident was 22.23 in the non-contested area and 356 in the contested area. Therefore, the use of cross-sectional variation is quite helpful in facilitating the discussion of the relationship between civil war and suicide.

## Graphical and descriptive statistical analysis

Fig 2 shows the average suicide rate in the pre-war and wartime periods in each district. In the pre-war period, the northern districts tend to exhibit higher suicide rate, suggesting a positive

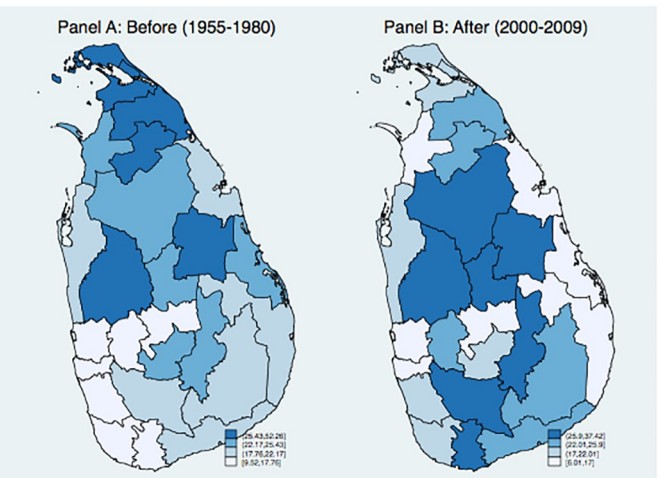

**Fig 2. The average suicide rate by districts.** The figures show the average suicide rate from 1955 to 1980 (Panel A) and from 2000 to 2009 (Panel B). (Source: [22, 23], *Statistics on Vital Events 2000–2010*).

correlation between the contested districts and suicide rate (Panel A). In contrast, during the wartime, the contested districts exhibit a lower suicide rate than the non-contested area (Panel B). Note that the suicide rate exhibits spatial clustering, the effects of which will be examined in the sensitivity analysis.

Fig 3 displays a much clearer contrast in the trend of average suicide rates between the contested and the non-contested districts: it was higher in the contested districts than in the non-contested area during the pre-war period, whereas the trend became totally converse during the wartime. The suicide rate in the non-contested area has a downward trend in the wartime period, and it is exceeded by the rate in the contested area in the post-war period in 2013.

The contrast between the contested and the non-contested districts can also be confirmed by the descriptive statistics (Table 1). In the contested districts, the suicide rate decreased by

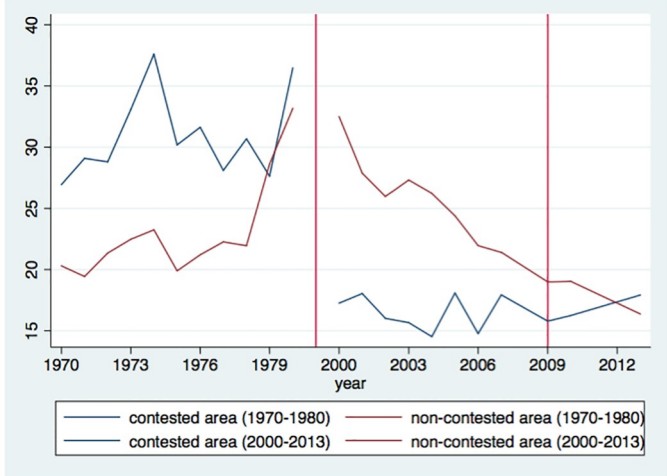

**Fig 3. Trends in suicide rate of contested and non-contested districts.** The figure shows the trend of average suicide rate in the pre-war (1970–1980), the war (2000–2009) and the post-war (2010 and 2013) periods for the contested and non-contested districts. (Source: [22, 23], *Statistics on Vital Events 2000–2010* and *Vital Statistics Report*).

**Table 1. Suicide rates (pre-war vs. wartime periods, contested vs. non-contested districts).**

|  | Pre-war period (1955–1980) | | | Wartime period (2000–2009) | | | Post-war period (2010 and 2013) | | |
|---|---|---|---|---|---|---|---|---|---|
| Contested districts | Count | Mean | SD | Count | Mean | SD | Count | Mean | SD |
| Total suicide (per 100,000) | 96 | 27.45 | 15.21 | 63 | 16.45 | 6.92 | 14 | 17.08 | 4.85 |
| Male suicide (per 100,000) | 48 | 34.82 | 21.18 | 63 | 23.40 | 10.12 | 14 | 25.14 | 7.47 |
| Female suicide (per 100,000) | 48 | 19.33 | 14.28 | 63 | 9.44 | 4.90 | 14 | 9.34 | 3.83 |
| Non-contested districts | Count | Mean | SD | Count | Mean | SD | Count | Mean | SD |
| Total suicide (per 100,000) | 207 | 20.63 | 10.63 | 135 | 25.18 | 8.86 | 30 | 17.70 | 4.60 |
| Male suicide (per 100,000) | 103 | 27.36 | 14.21 | 135 | 39.37 | 13.31 | 30 | 27.37 | 6.81 |
| Female suicide (per 100,000) | 103 | 13.25 | 8.38 | 135 | 10.97 | 4.93 | 30 | 8.36 | 3.87 |

The table demonstrates the summary statistics of suicide rates (per 100,000) for the contested and the non-contested districts as well as the pre-war, the wartime, and the post-war periods. (Source: [22, 23], *Statistics on Vital Events 2000–2010* and *Vital Statistics Report*).

11.0 points from the pre-war to the wartime period. In contrast, it increased by 4.55 points in the non-contested districts in the same period. Taking a difference-in-difference between the contested and the non-contested districts and between the pre-war and the wartime periods, the civil war lowered the suicide rate by 15.55 points. Similarly, the calculated effects for male and female suicide rates are 23.43 and 7.61 points, respectively.

Because of the data availability, the suicide rate discussed here focuses on a snapshot of the prolonged civil war, covering only the later period of the war. Thus, it is informative to overview the pattern of violence in the long run. Fig 4 shows that the number of grievous hurts stays more or less stable around 2,000 cases during wartime, which is lesser than that during the pre-war period. In contrast, the number of homicides has a general increasing trend from the pre-war to the wartime period. However, except for the sudden increase in homicides in 1988 and 1989, there is no clear trend in the pattern of violence during the wartime period. Thus, although the data during wartime should be interpreted with caution, our sample period

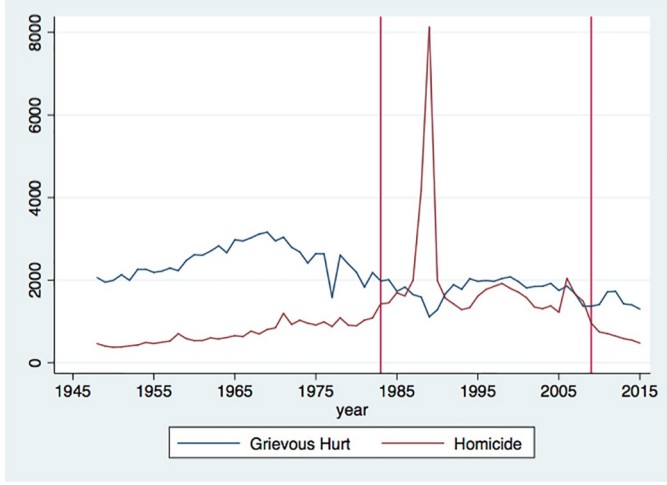

**Fig 4. Trends of grievous hurts and homicide at the national level.** The figure shows the trend of grievous hurts and homicide at the national level. It includes the pre-war (1948–1982), the war (1983–2009), and the post-war (2010–2015) periods. (Source: *Statistical Abstract*).

(2000–2009) is not necessarily a particular period of the civil war in terms of the pattern of violence.

These simple analyses confirmed the negative association between suicide rate and the Sri Lankan civil war. However, the analyses above were unconditional on any covariates, which makes it difficult to interpret the calculated difference as it is. It is essential to employ a regression approach to control these factors to obtain more rigorous evidence.

### Regression analysis

This study tested the relationship between the suicide rate and the Sri Lankan civil war by estimating the following regression model:

$$y_{it} = \beta D_i \cdot I(1983 \leq year_t \leq 2009) + X_{it}\gamma + \mu_i + \eta_t + \epsilon_{it},$$

where $y_{it}$ is suicide rate in district $i$ at time $t$; $D_i$ is an indicator variable that takes one if the district $i$ is occupied by LTTE and takes zero otherwise; $I(1983 \leq year_t \leq 2009)$ is an indicator of whether year $t$ corresponds to the conflict period (i.e., from 1983 to 2009); $X_{it}$ is the set of variables capturing other socio-economic conditions; and $\mu_i$ and $\eta_t$ are district and year fixed effects, respectively. By including these fixed effects, we controlled for district-specific heterogeneities and macro-level trend effect. It must be noted that this approach is a more flexible method to control for the trend effect than assuming a linear trend, which can be an important confounding factor. Also note that the year fixed effects control for the mean difference in the suicide rate from two different data sources. The parameter of interest is $\beta$, which represents the difference between the peacetime and the wartime periods as well as the difference between contested and non-contested areas. The standard errors were clustered at the district level.

We included several covariates that have been considered as significant predictors of the suicide rate in addition to the primary parameter of interest. Demographic variables such as population, population density (population per $km^2$), and sex ratio to capture the effect of internal migration [23] and the number of marriages [22] were included. The numbers of pupils per teacher and per school were included to capture the effect of educational availability [22]. As an economic indicator, the yield of paddy (kg/ha) in the main cropping season (*Maha*) was included as the share of the agricultural population is high and paddy is the main crop in the country. The number of deaths and infant mortality (per 1000 live births) were also included in the covariates to control for the public health conditions. The summary statistics of these variables are shown in Table 2. The map representations are also available in the supporting information (S1–S8 Figs). However, several essential variables such as unemployment rate could not be included in the regression model as available data at the district level is very limited. Therefore, several potential threats to identification are discussed in the following sections.

## Results

### Main analysis

Table 3 demonstrates the estimation results. First two columns show that the total suicide rate significantly decreased in the contested districts by 11.84–14.38 points (95% CI 6.46–17.22 and 7.21–21.54, respectively) during the wartime, which is almost comparable to the effect calculated from the summary statistics. Note that the variable of interest is a dummy, and the point estimate directly translated into the equivalent change in the suicide rate. Compared to the average suicide rate in the conflict districts during the pre-war period (27.5 per 100,000 population), these estimates correspond to 43–52% decrease, and thus the impact is psychiatrically,

**Table 2. Summary statistics of the covariates.**

| | Unit | Count | Mean | SD |
|---|---|---|---|---|
| Log (deaths) | Log | 506 | 8.12 | 0.90 |
| Log (marriages) | Log | 527 | 8.37 | 0.92 |
| Log (population) | Log | 546 | 13.17 | 0.82 |
| Male-female ratio | Ratio | 544 | 1.05 | 0.09 |
| Log (yield in *Maha*) | Log | 544 | 7.96 | 0.37 |
| Number of pupils per teacher | Number | 526 | 0.04 | 0.01 |
| Number of pupils per school | Number | 526 | 327.06 | 107.66 |
| Population density | per $km^2$ | 547 | 336.08 | 433.88 |
| Infant mortality | per 1,000 | 504 | 26.46 | 20.71 |

The table demonstrates the summary statistics of the covariates included in the regression analysis. (Source: *Statistical Abstract*).

**Table 3. Difference-in-difference approach to the relationship between the civil war and suicide rate (main analysis).**

| VARIABLES | (1) | (2) | (3) | (4) | (5) | (6) |
|---|---|---|---|---|---|---|
| | Total | Total | Male | Male | Female | Female |
| Conflict area x wartime | -14.38*** | -11.84*** | -21.06*** | -19.06*** | -6.31** | -4.63** |
| | (3.44) | (2.59) | (4.21) | (3.83) | (2.43) | (2.12) |
| Log (deaths) | | -2.85 | | -4.43 | | -1.36 |
| | | (4.51) | | (5.94) | | (2.90) |
| Log (marriages) | | 5.63** | | 6.57** | | 5.69*** |
| | | (2.25) | | (2.67) | | (2.01) |
| Log (population) | | -6.70 | | -7.65 | | -8.95** |
| | | (5.08) | | (8.20) | | (3.24) |
| Male-female ratio | | 45.68** | | 39.46 | | 29.42** |
| | | (18.84) | | (24.86) | | (12.61) |
| Log (yield in Maha) | | 2.81 | | 6.42* | | -0.75 |
| | | (2.49) | | (3.60) | | (2.07) |
| Number of pupils per teacher | | 269.70*** | | 435.40** | | 58.16 |
| | | (92.17) | | (155.10) | | (97.99) |
| Number of pupils per school | | 0.05*** | | 0.06* | | 0.03*** |
| | | (0.02) | | (0.03) | | (0.01) |
| Population density | | 0.00* | | 0.00 | | 0.00* |
| | | (0.00) | | (0.00) | | (0.00) |
| Infant mortality | | 0.03 | | 0.06 | | 0.01 |
| | | (0.06) | | (0.13) | | (0.05) |
| Year FE | YES | YES | YES | YES | YES | YES |
| District FE | YES | YES | YES | YES | YES | YES |
| Observations | 545 | 504 | 393 | 373 | 393 | 373 |
| R2 | .69 | .74 | .70 | .73 | .65 | .69 |

The table shows the effect of the civil war on the suicide rate, accounting for year and district fixed effects, as well as other covariates. The dependent variable is the suicide rate (per 100,000 population) of total population (columns 1 and 2), male population (columns 3 and 4), and female population (columns 5 and 6). Clustered standard errors at district level are reported in parentheses. (Source: Author's calculation).

as well as statistically, significant. It must be noted that this decrease is robust even to the inclusion of other controlling variables. Regarding the covariates, the only significant ones are the sex ratio and education variables (i.e., the number of pupils per teacher and per school). However, these variables are included for the precision of the parameter of interest and should not be interpreted as causal.

The negative association between suicide and the civil war remained for the suicide rates by gender, and the magnitude of the coefficient is much larger for male than female: 19.06–21.06 point (95% CI 11.09–27.03 and 12.31–29.81, respectively) decrease in male suicide and 4.63–6.31 point (95% CI 0.22–9.04 and 1.26–11.36, respectively) decrease in female suicide. Compared to the baseline average, these estimates translated to 55–61% and 24–33% decrease, respectively. These results suggest that male suicide is more responsive to the civil war.

## Sensitivity analysis

We conducted several sensitivity analyses to test the robustness of the negative association between the suicide rate and the civil war. Firstly, the above analysis defined the 2000–2009 period as the wartime. However, GOSL and LTTE signed the ceasefire agreement in 2002, and the peace process continued until 2006 when the Eelam War IV broke out. Therefore, it is informative to test whether the qualitative results change if we re-define the wartime, excluding the ceasefire period, and re-estimate the regression model.

Table 4 demonstrates the estimation results taking into account the ceasefire period. Although the magnitude of the coefficients becomes smaller than in Table 3, the point estimates are significantly negative except for the female suicide rate with covariates: the suicide rate declined by 4.83–9.96 points (CI 0.94–8.73 and 4.41–15.52, respectively) for the total population, 5.37–11.46 points (CI 0.37–10.37 and 5.28–17.64, respectively) for male, and 0.92–3.71 points (CI -1.29–3.12 and 0.43–6.98, respectively) for female. The decline in the magnitude may imply that the prolonged exposure to the civil war leads to a lower suicide rate.

The second concern in the main analysis is data reliability: It may be possible that the suicide rate data is less accurate in the contested area. Although we cannot directly test this issue, it is possible to re-estimate the model excluding the observations of the contested districts where data collection is expected to be difficult. Regarding this matter, the enumeration for the 2001 census was not conducted, albeit not entirely, in the contested districts with higher intensity of conflicts. Thus, it may be possible that the suicide statistics in these area during the sample period are less precise than those in other area or period.

Table 5 shows the estimation results without the observations for the districts where the 2001 census was not conducted. The point estimates still demonstrated a significantly negative relationship between the suicide rate and the civil war except for female suicides: the suicide rate in the remaining contested area declined by 4.98–6.93 points (CI 2.48–7.49 and 2.63–11.22, respectively) for the total population, 10.19–13.68 points (CI 5.76–14.62 and 5.49–21.87, respectively) for male, and 0.82–1.64 points (CI -5.76–2.21 and -1.45–4.74, respectively) for female. Note that the dropped districts have a higher intensity of the conflicts and are expected to have lower suicide rates according to Durkheim's theory. Thus, it is consistent that the point estimates become smaller than in Table 3.

The third possible concern about the robustness of the findings is that the estimates so far do not reflect the intensity of conflicts. Even within the contested districts, the intensity of conflicts—such as the number of the victims—varied substantially. Besides, several sporadic fights broke out even in the non-contested area during the wartime, implying that the Stable Unit Treatment Value Assumption (SUTVA) does not strictly hold in the current analysis. We re-estimated the model—to take into account these issues—using the number of victims from the

**Table 4. Difference-in-difference approach to the relationship between the civil war and suicide rate (the ceasefire agreement in 2002).**

| VARIABLES | (1) | (2) | (3) | (4) | (5) | (6) |
|---|---|---|---|---|---|---|
| | Total | Total | Male | Male | Female | Female |
| Conflict area x wartime (except for 2002–2006) | -9.96*** | -4.83** | -11.46*** | -5.37** | -3.71** | -0.92 |
| | (2.67) | (1.87) | (2.97) | (2.40) | (1.58) | (1.06) |
| Log (deaths) | | -1.07 | | -1.95 | | -0.74 |
| | | (5.14) | | (6.34) | | (2.94) |
| Log (marriages) | | 6.65** | | 9.22** | | 6.46*** |
| | | (2.91) | | (3.36) | | (2.26) |
| Log (population) | | -11.32 | | -14.58 | | -10.78** |
| | | (6.93) | | (11.69) | | (4.10) |
| Male-female ratio | | 48.87** | | 45.07 | | 30.93** |
| | | (21.17) | | (29.45) | | (13.71) |
| Log (yield in Maha) | | 1.12 | | 2.39 | | -1.75 |
| | | (2.72) | | (3.96) | | (2.10) |
| Number of pupils per teacher | | 468.89*** | | 807.58*** | | 151.12* |
| | | (120.83) | | (216.92) | | (82.63) |
| Number of pupils per school | | 0.05** | | 0.07* | | 0.04*** |
| | | (0.02) | | (0.03) | | (0.01) |
| Population density | | 0.01*** | | 0.01 | | 0.00*** |
| | | (0.00) | | (0.01) | | (0.00) |
| Infant mortality | | -0.03 | | -0.04 | | -0.02 |
| | | (0.08) | | (0.17) | | (0.06) |
| Year FE | YES | YES | YES | YES | YES | YES |
| District FE | YES | YES | YES | YES | YES | YES |
| Observations | 545 | 504 | 393 | 373 | 393 | 373 |
| R2 | .63 | .70 | .62 | .67 | .63 | .68 |

The table shows the effect of the civil war on the suicide rate, accounting for year and district fixed effects, as well as other covariates. The dependent variable is the suicide rate (per 100,000 population) of total population (columns 1 and 2), male population (columns 3 and 4), and female population (columns 5 and 6). Clustered standard errors at district level are reported in parentheses. (Source: Author's calculation).

conflicts instead of the variable of primary interest: the cross term of the contested area and the wartime dummy.

Table 6 shows the estimation results using the number of victims from the conflicts. The significantly negative association between the civil war and the suicide rate remained unchanged in this exercise. A one point increase in the number of deaths per 1,000 can be translated into the decline in suicide rate by 0.78–0.82 points (CI 0.15–1.42 and 0.57–1.08, respectively) for the total population, 0.71–0.84 points (CI 0.40–1.02 and 0.10–1.59, respectively) for male, and 0.45–0.58 (CI 0.14–0.76 and 0.49–0.68) for female. Because the average number of deaths from the conflicts per 1,000 population during the wartime is 0.81, its average impact on the suicide rate for the total population corresponds to 2.30–2.42% decline from the average in the pre-war period. The smaller magnitude of the coefficients than in previous tables suggests that the suicide rate is not so much responsive to the number of deaths in the conflict as to the fact that the place is in dispute.

The fourth possible concern is the spatial correlation in the suicide rate. As shown in Fig 2, the suicide rate exhibits spatial correlation, which can violate the identifying assumption of the difference-in-difference approach (i.e., SUTVA). For this reason, we first estimated the

**Table 5. Difference-in-difference approach to the relationship between the civil war and suicide rate (omitting districts where the enumeration was difficult).**

| VARIABLES | (1) | (2) | (3) | (4) | (5) | (6) |
|---|---|---|---|---|---|---|
| | Total | Total | Male | Male | Female | Female |
| Conflict area x wartime | -4.98*** | -6.93*** | -10.19*** | -13.68*** | -0.82 | -1.64 |
| | (1.20) | (2.07) | (2.13) | (3.94) | (0.67) | (1.49) |
| Log (deaths) | | -3.15 | | -7.24 | | -2.13 |
| | | (4.89) | | (6.97) | | (2.90) |
| Log (marriages) | | 3.12 | | 1.03 | | 5.60 |
| | | (2.40) | | (4.86) | | (4.54) |
| Log (population) | | -0.17 | | 7.62 | | -6.79 |
| | | (6.08) | | (13.99) | | (6.57) |
| Male-female ratio | | 49.11** | | 60.76* | | 24.07 |
| | | (23.54) | | (31.62) | | (17.15) |
| Log (yield in Maha) | | 3.16 | | 7.34** | | -0.61 |
| | | (2.43) | | (3.04) | | (2.56) |
| Number of pupils per teacher | | 146.96 | | 276.30 | | -65.04 |
| | | (126.21) | | (191.78) | | (166.90) |
| Number of pupils per school | | 0.03** | | 0.03 | | 0.03*** |
| | | (0.02) | | (0.03) | | (0.01) |
| Population density | | 0.00* | | 0.00 | | 0.00 |
| | | (0.00) | | (0.00) | | (0.00) |
| Infant mortality | | 0.04 | | 0.13 | | 0.03 |
| | | (0.06) | | (0.12) | | (0.04) |
| Year FE | YES | YES | YES | YES | YES | YES |
| District FE | YES | YES | YES | YES | YES | YES |
| Observations | 500 | 459 | 348 | 328 | 348 | 328 |
| R2 | .73 | .75 | .72 | .74 | .69 | .72 |

The table shows the effect of the civil war on the suicide rate, accounting for year and district fixed effects, as well as other covariates after dropping the observations of districts where the 2001 census was not carried out (i.e., Batticaloa, Jaffna, Kilinochchi, Mannar, Mullaitivu, Trincomalee, and Vavuniya). The dependent variable is the suicide rate (per 100,000 population) of total population (columns 1 and 2), male population (columns 3 and 4), and female population (columns 5 and 6). Clustered standard errors at district level are reported in parentheses. (Source: Author's calculation).

following spatial autoregressive (SAR) model with district fixed effects by the transformation approach to control the incidental parameter problem [30]:

$$y_{it} = \alpha \sum_j w_{ij} y_{jt} + \beta D_i \cdot I(1983 \leq year_t \leq 2009) + X_{it}\gamma + \mu_i + \eta_t + \epsilon_{it},$$

where $w_{ij}$ denotes the element of the adjacency matrix taking 1 if district $i$ and $j$ are adjacent and 0 otherwise. The weight matrix is row-standardized for the estimation.

Table 7 shows the estimation results of the SAR model. As expected, the spatial lag term is significantly positive for the total suicide rate. However, it lost statistical significance for the male suicide rate when additional control variables were included. As for the female suicide rate, it is insignificant whether we include additional control variables or not. Importantly, the main parameter of interest remains virtually unchanged from Table 3: the suicide rate in the contested area during wartime is lower by 11.27–13.14 points (CI 4.48–18.06 and 4.50–21.78, respectively) for the total population, 18.47–19.04 points (CI 8.35–28.58 and 8.64–29.43, respectively) for male, and 4.84–6.52 points (CI -0.29–9.96 and 0.51–12.54, respectively) for

**Table 6. Regression analysis of the relationship between the civil war and suicide rate (intensity of conflict).**

| VARIABLES | (1) | (2) | (3) | (4) | (5) | (6) |
|---|---|---|---|---|---|---|
| | Total | Total | Male | Male | Female | Female |
| # of death in conflict per population (1,000) | -0.82*** | -0.78** | -0.71*** | -0.84** | -0.58*** | -0.45*** |
| | (0.12) | (0.30) | (0.15) | (0.36) | (0.05) | (0.15) |
| Log (deaths) | | 1.25 | | 0.78 | | 0.63 |
| | | (4.97) | | (6.30) | | (2.69) |
| Log (marriages) | | 5.15* | | 7.17** | | 4.72** |
| | | (2.81) | | (3.23) | | (2.21) |
| Log (population) | | -11.61* | | -14.38 | | -9.93** |
| | | (6.45) | | (10.75) | | (3.80) |
| Male-female ratio | | 48.54** | | 44.62 | | 29.91** |
| | | (20.32) | | (28.48) | | (13.32) |
| Log (yield in Maha) | | 1.10 | | 2.38 | | -1.68 |
| | | (2.60) | | (3.91) | | (1.98) |
| Number of pupils per teacher | | 554.59*** | | 918.98*** | | 197.42** |
| | | (137.46) | | (235.42) | | (77.44) |
| Number of pupils per school | | 0.06** | | 0.08** | | 0.04*** |
| | | (0.02) | | (0.04) | | (0.01) |
| Population density | | 0.01*** | | 0.01 | | 0.00*** |
| | | (0.00) | | (0.01) | | (0.00) |
| Infant mortality | | -0.06 | | -0.07 | | -0.03 |
| | | (0.08) | | (0.17) | | (0.05) |
| Year FE | YES | YES | YES | YES | YES | YES |
| District FE | YES | YES | YES | YES | YES | YES |
| Observations | 545 | 504 | 393 | 373 | 393 | 373 |
| R2 | .62 | .70 | .60 | .67 | .63 | .69 |

The table shows the effect of the civil war on the suicide rate, accounting for year and district fixed effects, as well as other covariates. The dependent variable is the suicide rate (per 100,000 population) of total population (columns 1 and 2), male population (columns 3 and 4), and female population (columns 5 and 6). Clustered standard errors at district level are reported in parentheses. (Source: Author's calculation).

female. Note that the sample size reduces from Table 3 because the estimation of the SAR model requires a strictly balanced panel data, and we drop observations with missing values.

As a next exercise, we estimated the following spatial difference-in-difference approach to test the spillover effect of civil war from the contested to non-contested areas [31]:

$$y_{it} = \alpha \sum_j w_{ij} D_i \cdot I(1983 \leq year_t \leq 2009) + \beta D_i \cdot I(1983 \leq year_t \leq 2009) + X_{it}\gamma + \mu_i + \eta_t + \epsilon_{it}.$$

Table 8 shows the results of this approach. The spillover term is statistically insignificant in all specifications, suggesting that there is no spillover effect from contested to the non-contested area. Indeed, the estimated parameter of interest remains virtually unchanged from Table 3: the suicide rate decreased in the contested area during wartime by 11.74–14.58 points (CI 6.43–17.06 and 7.27–21.90, respectively) for the total population, 18.73–21.24 points (CI 11.07–26.40 and 12.31–30.17, respectively) for male, and 4.46–6.42 points (CI 0.21–8.71 and 1.33–11.50, respectively) for female. These exercises suggest that spatial factors are not so serious as to affect the estimated reduction in suicide rate in the contested area during wartime.

**Table 7. Difference-in-difference approach to the relationship between the civil war and suicide rate (spatial autoregressive model).**

| VARIABLES | (1) | (2) | (3) | (4) | (5) | (6) |
|---|---|---|---|---|---|---|
| | Total | Total | Male | Male | Female | Female |
| Spatial lag term | 0.19** | 0.15* | 0.20** | 0.17 | 0.04 | 0.01 |
| | (0.08) | (0.08) | (0.09) | (0.11) | (0.09) | (0.07) |
| Conflict area x wartime | -13.14*** | -11.27*** | -19.04*** | -18.47*** | -6.52** | -4.84* |
| | (4.41) | (3.46) | (5.30) | (5.16) | (3.07) | (2.61) |
| Log (deaths) | | -2.01 | | -3.16 | | -0.62 |
| | | (3.82) | | (4.50) | | (2.69) |
| Log (marriages) | | 4.58** | | 2.47 | | 5.09*** |
| | | (1.89) | | (2.22) | | (1.94) |
| Log (population) | | -8.89* | | -8.21 | | -9.25*** |
| | | (5.36) | | (9.09) | | (3.54) |
| Male-female ratio | | 40.22** | | 23.47 | | 33.69** |
| | | (19.66) | | (26.15) | | (14.39) |
| Log (yield in Maha) | | 4.21* | | 7.53** | | -0.82 |
| | | (2.41) | | (3.48) | | (2.46) |
| Number of pupils per teacher | | 179.52* | | 340.46** | | -11.17 |
| | | (103.03) | | (164.09) | | (124.10) |
| Number of pupils per school | | 0.04*** | | 0.05* | | 0.03*** |
| | | (0.02) | | (0.03) | | (0.01) |
| Population density | | 0.00 | | -0.00 | | 0.00 |
| | | (0.00) | | (0.00) | | (0.00) |
| Infant mortality | | 0.04 | | 0.09 | | 0.00 |
| | | (0.06) | | (0.09) | | (0.05) |
| Year FE | YES | YES | YES | YES | YES | YES |
| District FE | YES | YES | YES | YES | YES | YES |
| Observations | 374 | 374 | 286 | 286 | 286 | 286 |
| R2 | .16 | .21 | .23 | .19 | .27 | .30 |

The table shows the effect of the civil war on the suicide rate, accounting for spatial dependence in the dependent variable, year and district fixed effects, as well as other covariates. The dependent variable is the suicide rate (per 100,000 population) of total population (columns 1 and 2), male population (columns 3 and 4), and female population (columns 5 and 6). Robust standard errors at district level are reported in parentheses. (Source: Author's calculation).

## Remaining threats to identification

We have revealed the negative relationship between suicide rate and the civil war, which is robust to several sensitivity analyses. The fundamental assumption was that there were no time-variant heterogeneities that are not parallel between the contested and the non-contested districts. However, we cannot reject the possibility that the assumption was not strictly applied in the current analysis. Notably, several important factors to be controlled were treated as the error term due to the limitation of data availability. Thus, it is useful to discuss the potential concern arising from omitting several important variables: economic indicators, pesticide regulation, and ethnicity.

Economic indicators such as GDP and unemployment rate are known as important factors to predict suicide rate [32–35]: a lower GDP and higher unemployment rate are expected to lead to higher suicide rate. Therefore, the correlation between these economic indicators and the civil war leads to biased estimates of the main parameter of interest. Related to this issue, a previous study presented that the decrease in the suicide rate in the US during wartime was spurious because of the decline in the unemployment rate during wartime [15]. However, in general, civil conflicts are considered to be associated with a lower GDP and higher

**Table 8. Spatial difference-in-difference approach to the relationship between the civil war and suicide rate.**

| VARIABLES | (1) | (2) | (3) | (4) | (5) | (6) |
|---|---|---|---|---|---|---|
| | Total | Total | Male | Male | Female | Female |
| Conflict area x wartime | -14.58*** | -11.74*** | -21.24*** | -18.73*** | -6.42** | -4.46** |
| | (3.52) | (2.56) | (4.29) | (3.69) | (2.44) | (2.04) |
| Spatial term of conflict area x wartime | 4.40 | 5.65 | 4.56 | 7.81 | 2.75 | 4.01 |
| | (4.56) | (3.51) | (6.09) | (5.92) | (3.80) | (3.39) |
| Log (deaths) | | -2.45 | | -4.12 | | -1.20 |
| | | (4.29) | | (5.67) | | (2.80) |
| Log (marriages) | | 5.98** | | 7.35** | | 6.09*** |
| | | (2.41) | | (3.03) | | (2.00) |
| Log (population) | | -7.65 | | -8.94 | | -9.61*** |
| | | (5.11) | | (8.33) | | (3.05) |
| Male-female ratio | | 45.05** | | 39.42 | | 29.40** |
| | | (17.82) | | (23.58) | | (12.01) |
| Log (yield in Maha) | | 3.28 | | 7.29** | | -0.30 |
| | | (2.49) | | (3.41) | | (2.46) |
| Number of pupils per teacher | | 274.04** | | 455.49*** | | 68.49 |
| | | (96.86) | | (158.88) | | (95.85) |
| Number of pupils per school | | 0.05*** | | 0.06* | | 0.03*** |
| | | (0.02) | | (0.03) | | (0.01) |
| Population density | | 0.01** | | 0.01 | | 0.00** |
| | | (0.00) | | (0.01) | | (0.00) |
| Infant mortality | | 0.03 | | 0.06 | | 0.01 |
| | | (0.06) | | (0.13) | | (0.05) |
| Year FE | YES | YES | YES | YES | YES | YES |
| District FE | YES | YES | YES | YES | YES | YES |
| Observations | 545 | 504 | 393 | 373 | 393 | 373 |
| R2 | .69 | .74 | .70 | .73 | .65 | .70 |

The table shows the effect of ethinicity on the suicide rate, accounting for spatial term of the main variable, year and district fixed effects, as well as other covariates. The dependent variable is the suicide rate (per 100,000 population) of total population (columns 1 and 2), male population (columns 3 and 4), and female population (columns 5 and 6). Clustered standard errors at district level are reported in parentheses. (Source: Author's calculation).

unemployment rate [36]. In the case of Sri Lanka, the unemployment rates in the Northern and Eastern Provinces—corresponding to the contested district—were much higher than the national average in 2003 [37]. Subsequently, these omitted variables are expected to cause upward bias in the estimated coefficient of the impact of civil war, resulting in the attenuation of the negative impact. However, even these conservative estimates were significantly negative, implying the strong negative impact of civil conflict on the suicide rate.

Another important omitted factor is pesticide banning. Pesticide poisoning has been the most common method of suicide in Sri Lanka, and the restrictions on the import and sales of highly toxic pesticide coincided with the reduction in suicides in the country [18, 38–40]. However, these pesticide regulations should be regarded as macro shocks that affect all the districts. By taking a difference-in-difference between the peacetime and the wartime periods as well as contested and non-contested districts, the estimated parameter is robust to these macro shocks. Even if the enforcement of the regulations was weak in the contested area, it only results in upward bias in the estimated coefficient and does not alter the interpretation of the main result. Therefore, pesticide restriction cannot explain the estimated negative relationship

**Table 9. Regression analysis of the relationship between the civil war and suicide rate (ethnicity).**

| VARIABLES | (1) | (2) | (3) | (4) | (5) | (6) |
|---|---|---|---|---|---|---|
|  | Total | Total | Male | Male | Female | Female |
| Share of Sri Lankan Tamil population | 12.01 | 127.99 | 0.25 | 75.92 | 9.93 | 41.33 |
|  | (20.75) | (171.69) | (32.01) | (305.56) | (7.53) | (118.35) |
| Share of Indian Tamil population | 38.39 | 66.26 | 68.30 | 104.95 | -0.45 | 32.92 |
|  | (31.33) | (88.51) | (50.57) | (97.07) | (14.78) | (51.84) |
| Log (deaths) |  | -1.56 |  | 9.51 |  | -9.94** |
|  |  | (32.99) |  | (10.79) |  | (4.11) |
| Log (marriages) |  | 9.03 |  | 11.03 |  | 3.83 |
|  |  | (8.61) |  | (10.88) |  | (4.33) |
| Log (population) |  | -20.90 |  | -37.05 |  | -7.06 |
|  |  | (22.60) |  | (43.82) |  | (16.18) |
| Male-female ratio |  | -54.73 |  | -73.46 |  | -33.97 |
|  |  | (179.20) |  | (183.81) |  | (72.55) |
| Log (yield in Maha) |  | 6.32 |  | 7.70 |  | -4.75 |
|  |  | (12.53) |  | (15.29) |  | (6.18) |
| Number of pupils per teacher |  | 21.34 |  | 362.77 |  | 146.30 |
|  |  | (572.05) |  | (616.89) |  | (363.25) |
| Number of pupils per school |  | 0.00 |  | 0.04 |  | 0.04 |
|  |  | (0.04) |  | (0.06) |  | (0.03) |
| Population density |  | -0.00 |  | -0.01 |  | -0.00 |
|  |  | (0.00) |  | (0.01) |  | (0.00) |
| Infant mortality |  | 0.09 |  | -0.17 |  | 0.11 |
|  |  | (0.52) |  | (0.66) |  | (0.22) |
| Year FE | YES | YES | YES | YES | YES | YES |
| District FE | YES | YES | YES | YES | YES | YES |
| Observations | 80 | 61 | 80 | 61 | 80 | 61 |
| R2 | .79 | .80 | .79 | .80 | .78 | .78 |

The table displays the effect of the civil war on the suicide rate, accounting for year and district fixed effects, as well as other covariates. The samples are restricted to the census years (i.e., 1955, 1965, 1971, and 2001). The dependent variable is the suicide rate (per 100,000 population) of total population (columns 1 and 2), male population (columns 3 and 4), and female population (columns 5 and 6). Indian Tamils are Tamil people of Indian origin, while Sri Lankan Tamils are native to Sri Lanka. Clustered standard errors at district level are reported in parentheses. (Source: Author's calculation).

between suicide and the civil war. Notably, this does not contradict the argument that the restriction of toxic pesticide led to the reduction in suicide; the civil war, as well as pesticide reduction, led to the decrease in suicide.

Ethnicity is also considered to be an important factor leading to suicide. In the Sri Lankan context, the Tamils have a higher suicide rate [22], and the contested districts were associated with a significantly higher share of Tamil population. Thus, the estimated coefficient might confound with ethnicity. The straightforward way is to include the share of the Tamil population in the regression model to control for the effect of ethnicity, but this information is only available for the year when the census was conducted. Besides, more importantly, the data is missing for the conflict area during wartime due to non-implementation of the enumeration, which makes it impossible to estimate the same specification as before. However, it is still possible to test the effect of ethnicity by estimating more parsimonious specification using only available observations.

Table 9 shows the association between the suicide rate and ethnicity using the observations in census years (i.e., 1955, 1965, 1971, and 2001). After the control of district and year-specific

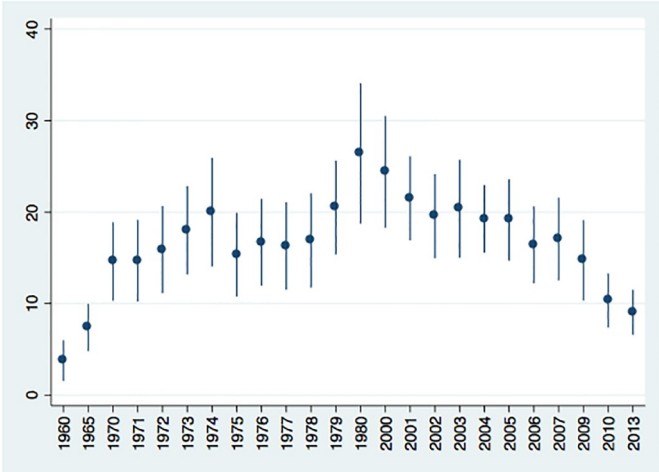

**Fig 5. Trend in suicide rate adjusted for the effect of the civil war and district-specific heterogeneities.** The figure shows the potential suicide rate measured as the estimates of year fixed effects of (1) in Table 3. The suicide rate in 1955 is set to be zero. (Source: author's calculation).

heterogeneities, the estimated results disclosed that the Tamil population was not necessarily more likely to commit suicide. Therefore, there is no sound reason to believe that the estimated impact was confounded with the effect of ethnicity. Furthermore, if the Tamils had higher suicide rate and the population of these people was high in the contested area, this only resulted in the upward bias in the estimated coefficient, which does not do away the negative relationship between suicide and the civil war.

## Discussion

We have demonstrated the decrease in annual suicide rate in Sri Lanka during the civil war by estimating the difference-in-difference between the pre-war and wartime periods and between the contested and non-contested districts. The finding was robust to several sensitivity analyses. We also discussed that several confounding factors that cannot be incorporated in the regression model did not necessarily affect the finding.

There has been no consensus on whether the civil war resulted in a lower suicide rate in Sri Lanka: The decline was confirmed within a district while it was not confirmed at the national level. These conflicting views stem from analysing only time-series variations. Thus, it is informative to discuss the trend in the suicide rate after controlling for time-invariant district level heterogeneities and potential suicide rate in each district after controlling for year-specific heterogeneities in order to highlight this issue. First, the average level of suicide rate did increase during the wartime even after controlling for the district-level heterogeneities, contrasting the negative relationship found in the analyses (Fig 5). Notably, the trend is consistent with the one reported by a previous study [18]. Second, the contested areas tended to have a higher potential suicide rate after controlling for the year-specific heterogeneities, which also masks the decrease in the suicide rate during wartime (Fig 6). Therefore, exploiting only time-series or cross-sectional variation is not sufficient to accurately detect the decline in the suicide rate during wartime, which strengthens the advantage of the current analysis.

An important limitation in the current analysis is that the suicide rate data at the district-level were not fully available, especially in the early stage of the civil war. The annual suicide rate in Sri Lanka increased drastically at the end of the 1970s and during the early 80s,

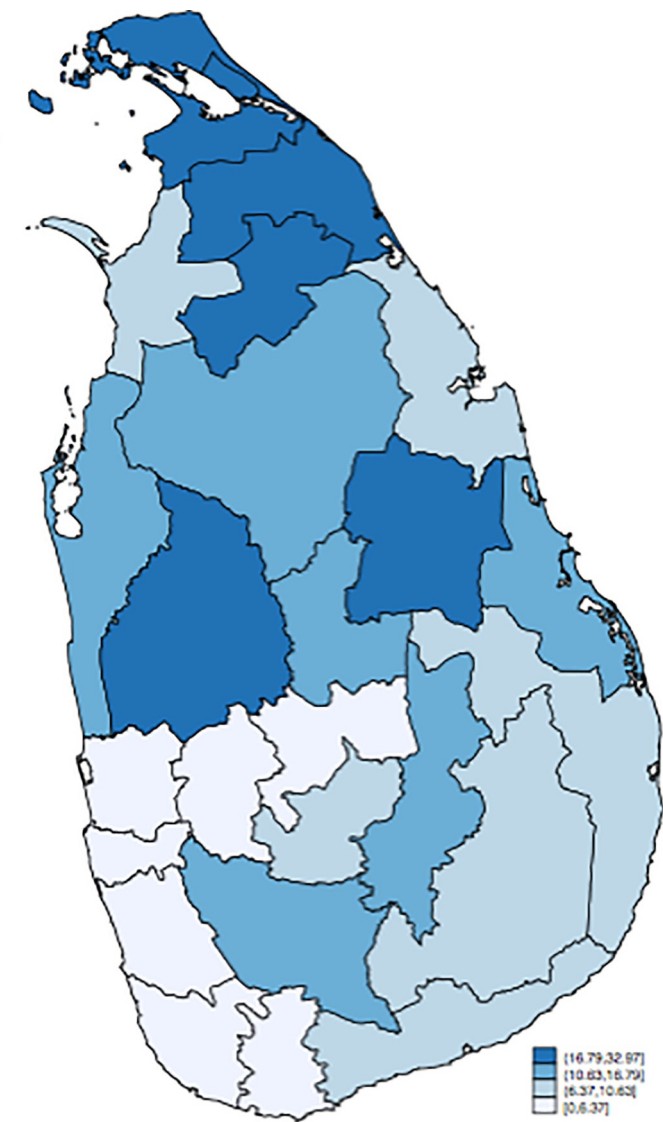

**Fig 6. Suicide rate adjusted for the effect of the civil war and year-specific heterogeneities.** The figure shows the potential suicide rate measured as the estimates of district fixed effects of (1) in Table 3. The suicide rate in Colombo is set to be zero. (Source: Author's calculation).

continuing to remain high until 1995, and started to decrease since then [18, 19]. Therefore, the decline in the suicide rate during wartime was confirmed only for the late stage of the civil war when the suicide rate had already been in the decreasing trend. Moreover, the analysis covered only for two years in the post-war period, which is not sufficient to discuss the trend after the civil war. Although these limitations do not affect the finding of this study, it is desirable to estimate the same regression model using the dataset covering the whole wartime and post-war periods to obtain a more comprehensive understanding of the issue. Thus, extending the data remains an important future task.

Another possible limitation is that the current study did not analyze the mechanism per se to explain why the civil war led to a lower suicide rate though it is not necessarily the scope of this study. We have discussed that the decline in the suicide rate during wartime is not spurious. Thus, the finding can be interpreted in the light of the Durkheimian theory, which places

importance on social integration as a determinant of suicide [1, 41]. As mentioned, several studies in comparative politics have argued that civil wars are associated with the construction of ethnic identity [9–11], which can lead to lowering of the suicide rate. In fact, there is a report summarizing—in the case of the Sri Lankan civil war—such association especially among the youth [42]. However, further in-depth analysis is required to test the validity of this mechanism.

## Supporting information

**S1 Fig. Heatmap of log (# of death).** The figures show the average of the log (# of death) from 1970 to 1980 (Panel A) and from 2000 to 2013 (Panel B).
(TIF)

**S2 Fig. Heatmap of log (# of marriage).** The figures show the average of the log (# of marriage) from 1970 to 1980 (Panel A) and from 2000 to 2013 (Panel B).
(TIF)

**S3 Fig. Heatmap of log (population).** The figures show the average of the log (population) from 1970 to 1980 (Panel A) and from 2000 to 2013 (Panel B).
(TIF)

**S4 Fig. Heatmap of male-female ratio.** The figures show the average of the male-female Ratio from 1970 to 1980 (Panel A) and from 2000 to 2013 (Panel B).
(TIF)

**S5 Fig. Heatmap of log (yield in Maha).** The figures show the average of the log (yield in Maha) from 1970 to 1980 (Panel A) and from 2000 to 2013 (Panel B).
(TIF)

**S6 Fig. Heatmap of # of pupils per school.** The figures show the average of the # of pupils per school from 1970 to 1980 (Panel A) and from 2000 to 2013 (Panel B).
(TIF)

**S7 Fig. Heatmap of population density.** The figures show the average of the population density from 1970 to 1980 (Panel A) and from 2000 to 2013 (Panel B).
(TIF)

**S8 Fig. Heatmap of infant mortality.** The figures show the average of the infant mortality from 1970 to 1980 (Panel A) and from 2000 to 2013 (Panel B).
(TIF)

**S1 File.**
(ZIP)

## Acknowledgments

I am grateful to the editor, two anonymous referees, Kyoji Fukao, Kazunobu Hayakawa, Takayuki Higashikata, Masami Ishida, Takeshi Kawanaka, Nobuyoshi Kikuchi, Yuya Kudo, Tomohiro Machikita, and Momoe Makino for their constructive comments. All remaining errors are my own.

## Author Contributions

**Conceptualization:** Takeshi Aida.

**Data curation:** Takeshi Aida.

**Formal analysis:** Takeshi Aida.

**Funding acquisition:** Takeshi Aida.

**Investigation:** Takeshi Aida.

**Methodology:** Takeshi Aida.

**Project administration:** Takeshi Aida.

**Resources:** Takeshi Aida.

**Software:** Takeshi Aida.

**Supervision:** Takeshi Aida.

**Validation:** Takeshi Aida.

**Visualization:** Takeshi Aida.

**Writing – original draft:** Takeshi Aida.

**Writing – review & editing:** Takeshi Aida.

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
