## [Decision Letter · Decision Letter 0]

20 May 2020

PONE-D-20-10169

Revisiting suicide rate during wartime: Evidence from the Sri Lankan civil war

PLOS ONE

Dear Dr. Aida,

Thank you for submitting your manuscript to PLOS ONE. After careful consideration, we feel that it has merit but does not fully meet PLOS ONE’s publication criteria as it currently stands. Therefore, we invite you to submit a revised version of the manuscript that addresses the points raised during the review process.

Please carefully review the comments of both reviewers and myself, when revising your manuscript. Pay particular attention to the cultural and local views that are likely unique in Sri Lanka, and a critical approach to theory would be helpful. In addition, please clarify the statistical analyses and provide more details of the results.

We would appreciate receiving your revised manuscript by Jul 04 2020 11:59PM. To enhance the reproducibility of your results, we recommend that if applicable you deposit your laboratory protocols in protocols.io, where a protocol can be assigned its own identifier (DOI) such that it can be cited independently in the future. For instructions see: http://journals.plos.org/plosone/s/submission-guidelines#loc-laboratory-protocols

We look forward to receiving your revised manuscript.

Kind regards,

Keith M. Harris, PhD

Academic Editor

PLOS ONE

Additional Editor Comments:

Thank you for submitting this interesting work. Both reviewers found the topic important and of interest. Both also found significant limitations in the theoretical approach and evaluation of local cultural factors. I concur with Reviewer 2 that a major revision is appropriate. Please address the cultural factors, perhaps consulting Sri Lankan' sources on local views toward the civil war and suicide. Additional theoretical approaches should also be examined. This work may actually demonstrate limitations in Durkheim's theory, rather than support. That may be a contribution of this study. In addition, please address issues with the data. Table 2 requires dates for the pre-war and war periods. Also, why not include post-war data? Typically, we provide data at two decimal places, rather than three. Tables 4-8 require more clarification. Mention the type of analysis in the title, and more details of the variables, what do the numbers represent? If these are regression models, they require R-squared values and standardized beta values for better interpretation of the total model and individual predictors. Overall, the analyses and results require better elaboration and further interpretation.

2. Please include captions for your Supporting Information files at the end of your manuscript, and update any in-text citations to match accordingly. Please see our Supporting Information guidelines for more information: http://journals.plos.org/plosone/s/supporting-information

Reviewers' comments:

Reviewer's Responses to Questions

**Comments to the Author**

1. Is the manuscript technically sound, and do the data support the conclusions?

Reviewer #1: No

Reviewer #2: Yes

2. Has the statistical analysis been performed appropriately and rigorously? 

Reviewer #1: No

Reviewer #2: Yes

3. Have the authors made all data underlying the findings in their manuscript fully available?

Reviewer #1: No

Reviewer #2: Yes

4. Is the manuscript presented in an intelligible fashion and written in standard English?

Reviewer #1: Yes

Reviewer #2: Yes

5. Review Comments to the Author

Reviewer #1: The paper addresses an important issue, but I do not think the Sri Lankan civil war (1983 - 2009) provides suitable case study of the issue. This was a separatist was between LTTE (a group of radical youth from the second largest ethnic group, Tamils, of the country) and the government). There was no broad community support for the LTTE among in the Tamil community in the contested districts. Put simply, during this period certainly there was no 'increased social integration' that would have led to fewer suicide cases in these districts as postulated by Durkhein (1887).

Given this fundamental flaw, I do not which to make detailed comments on this paper.

Reviewer #2: The article under consideration investigates the relationship between war and suicide at the local-level using Sri Lanka as a case study. I find the topic important, and was impressed by the work the author did to cope with limited data on the topic. That said, I do have several concerns that I summarize below.

• The premise is the work of Durkheim, who argues that war increases social integration, thus leading to fewer suicides. The issue here is that the author does not really bring this logic into the article, so the reader is left wondering what the debate is about. I would like to see a clear theoretical discussion about how this logic works, and why specifically war is able to increase social integration.

It is worth noting that war has changed dramatically since 1897 (when Durkheim published this piece), so it is important for the author the explain the mechanisms by which the war in Sri Lanka might increase social integration. I think this argument is clear in the context of interstate wars, but not civil wars – especially one as long-lasting as Sri Lanka.

• The author also notes that the “other side” argues that confounding factors are driving this relationship, but fails to elaborate. What exactly does this side of the literature argue, and what confounding factors are seen as being most significant?

• In my view, the introduction could be improved by being explicit about the importance of the study. The author should look to frame their contribution in the larger body of work that looks at the consequences of civil wars, and the fact that these consequences linger long after the guns fall silent. Or, alternatively, the author could cite some literature that speaks to the societal and personal costs of suicide. But, in any case, some work needs to be done to improve this.

• Because the war in Sri Lanka was so prolonged, there is a need to discuss the evolution of the war. Specifically, I would like to see some data (or a discussion if data is not possible) that lays out patterns of violence over time. The empirical work here only looks at a brief snapshot of the war (which was during the final phase), so it would be useful to see how this period compares to others.

• A discussion of how the suicide rate was calculated in this case would be helpful. It is very difficult to get information during wartime, even battle deaths. This implies that any suicide rates reported during wartime should be interpreted with extreme caution.

How heavily is suicide stigmatized in Sri Lanka? Is there an incentive to add suicides to fatalities via the war rather than acknowledge suicide? How exactly does this process play out? I know some of these questions may not be answerable, but this has important implications for the negative relationship that is uncovered. I would like to see the author at least address this potential issue.

• Empirically, I have two concerns. First, I am not entirely sure where the control variables come from or how they are distributed across space. Because the study is done at the local-level it is not enough to show the aggregate distribution, the author must also show local variation. Creating some maps would be great here.

Second, there is notable spatial clustering in Figure 2. Has the author ruled out spatial dependence? Or, is there a need to account for diffusion in the models?

6. PLOS authors have the option to publish the peer review history of their article (what does this mean?). If published, this will include your full peer review and any attached files.

Reviewer #1: No

Reviewer #2: No

---

## [Author Response · Author response to Decision Letter 0]

3 Jul 2020

[Comments from the editor]

Thank you for submitting this interesting work. Both reviewers found the topic important and of interest. Both also found significant limitations in the theoretical approach and evaluation of local cultural factors. I concur with Reviewer 2 that a major revision is appropriate. Please address the cultural factors, perhaps consulting Sri Lankan' sources on local views toward the civil war and suicide. Additional theoretical approaches should also be examined. This work may actually demonstrate limitations in Durkheim's theory, rather than support. That may be a contribution of this study. In addition, please address issues with the data. Table 2 requires dates for the pre-war and war periods. Also, why not include post-war data? Typically, we provide data at two decimal places, rather than three. Tables 4-8 require more clarification. Mention the type of analysis in the title, and more details of the variables, what do the numbers represent? If these are regression models, they require R-squared values and standardized beta values for better interpretation of the total model and individual predictors. Overall, the analyses and results require better elaboration and further interpretation.

[Reply]

Thank you for your kind consideration and constructive comments. I hope the current version conforms to the standards of your journal.

- In accordance with the suggestions from you and Reviewer #2, I have included a discussion on the cultural factors surrounding the stigmatization of suicide in Sri Lanka.

- I want to thank you for this very important comment. Taking heed of your and the reviewers’ advice, I have elaborated on Durkheim’s theory and how civil wars can lead to lower suicide rates. Specifically, several studies in comparative politics have shown that civil wars harden ethnic identities. Thus, civil wars may boost the level of integration, although its body comprises ethnic groups rather than a nation, and may lower suicide rates.

- I have added the year in Table 2.

- I found a new data source and expanded my data to include the post-war period (years 2010 and 2013). Although it is still limited, it provides us some clues about the suicide rate in the post-war period. It is also mentioned in the discussion section that further expansion of the data is an important work that still remains unfinished.

- The data is now shown up to two decimal places.

- I have added a more detailed interpretation of the results shown in Tables 4–10.

- I have mentioned the type of analysis in the title of the tables and added the units of each variable in Table 3 as well as in the main text.

- R-squares are in the tables of the regression results (Tables 4–10). As for standardized beta values, I agree that they make the comparison of the coefficients easier in general. However, our primary variable of interest (that is, a cross-term of the wartime and contested area dummies) is a dummy variable. Therefore, the point estimate is directly translated into the change in the suicide rate compared to the non-contested area in the peacetime period. For this reason, I am worried that standardized beta values might make the interpretation more difficult. Additionally, I have not interpreted other coefficients because they are just control variables, and thus, this should not be interpreted as being causal. Therefore, I have kept the variables unstandardized and have mentioned these issues in the text. However, if you think that standardized beta values are better for the control variables, I would be happy to make changes accordingly.

\f

[Comments from Reviewer #1]

The paper addresses an important issue, but I do not think the Sri Lankan civil war (1983 - 2009) provides suitable case study of the issue. This was a separatist was between LTTE (a group of radical youth from the second largest ethnic group, Tamils, of the country) and the government). There was no broad community support for the LTTE among in the Tamil community in the contested districts. Put simply, during this period certainly there was no 'increased social integration' that would have led to fewer suicide cases in these districts as postulated by Durkhein (1887).

Given this fundamental flaw, I do not which to make detailed comments on this paper.

[Reply]

Thank you for pointing out a very critical issue. It is true that the Sri Lankan Civil War was fought between a group of separatists (LTTE) and the government, which is different from the wars discussed by Durkheim.

In order to incorporate the suggestions in your comment, as well as the one from Reviewer #2, I have rewritten the introduction. In the current version, I have elaborated on Durkheim’s theory and explained why it may be applicable to civil wars. Specifically, several studies in comparative politics have shown that civil wars harden ethnic identities. Thus, it is possible that civil wars enhance the level of integration, although its body consists of ethnic groups rather than a nation, and lead to lower suicide rates. I have mentioned this as an important issue remaining to be addressed in the introduction.

\f

[Comments from Reviewer #2]

The article under consideration investigates the relationship between war and suicide at the local-level using Sri Lanka as a case study. I find the topic important, and was impressed by the work the author did to cope with limited data on the topic. That said, I do have several concerns that I summarize below.

• The premise is the work of Durkheim, who argues that war increases social integration, thus leading to fewer suicides. The issue here is that the author does not really bring this logic into the article, so the reader is left wondering what the debate is about. I would like to see a clear theoretical discussion about how this logic works, and why specifically war is able to increase social integration.

[Reply]

I am much obliged for your very detailed and constructive comments. I have incorporated your suggestions to the best of my ability. I firmly believe that the manuscript has improved significantly.

In response to your observation, I have elaborated on Durkheim’s theory in the introduction. Specifically, he argues that wars “sharpen collective feelings, stimulate the party spirit and the national one and, by concentrating activities towards a single end, achieve, at least for a time, greater integration of society.” (Durkheim, 1897 [1]) Thus, it is understood that wars increase social integration, which leads to lower suicide rates.

[Comment]

It is worth noting that war has changed dramatically since 1897 (when Durkheim published this piece), so it is important for the author the explain the mechanisms by which the war in Sri Lanka might increase social integration. I think this argument is clear in the context of interstate wars, but not civil wars – especially one as long-lasting as Sri Lanka.

[Reply]

Thank you for pointing out a very critical issue. It is true that Durkheim considered inter-state wars in his argument while the war in Sri Lanka was a civil conflict, which was fought between the LTTE and the government. 

In accordance with your comment, as well as the one from Reviewer #1, I have rewritten the introduction. Specifically, several comparative political studies have shown that civil wars harden ethnic identities. Thus, it is possible that civil wars increase the level of integration, though its body encompasses ethnic groups rather than a nation, and lower suicide rates. I have mentioned this as an important issue remaining to be addressed in the introduction.

[Comment]

• The author also notes that the “other side” argues that confounding factors are driving this relationship, but fails to elaborate. What exactly does this side of the literature argue, and what confounding factors are seen as being most significant?

[Reply]

I have elaborated on the omitted confounding factors discussed in the previous studies in the revised introduction. Specifically, they are mainly economic conditions and time-trend. As for the economic conditions, I included the yield of paddy in the primary cropping season. Unfortunately, I could not include the unemployment rate because of the non-availability of data. However, I have discussed why the unemployment rate is not a severe concern in this study. As for the trend effect, the difference-in-difference approach with year fixed effects is a more flexible way to control it than including the linear trend effect. For these reasons, these factors are not so critical as to change the findings of this study.

[Comment]

• In my view, the introduction could be improved by being explicit about the importance of the study. The author should look to frame their contribution in the larger body of work that looks at the consequences of civil wars, and the fact that these consequences linger long after the guns fall silent. Or, alternatively, the author could cite some literature that speaks to the societal and personal costs of suicide. But, in any case, some work needs to be done to improve this.

[Reply]

I appreciate this extremely significant recommendation. As mentioned above, I have extensively rewritten the introduction following your advice. In the revised version, I have explained that the applicability of Durkheim’s theory in the context of civil wars is an important issue remaining to be addressed. In the discussion, I have mentioned that the existing studies have mainly focused on the long-lasting effect of civil wars on physique and diseases, but the issue of wartime suicides has been largely unexplored.

[Comment]

• Because the war in Sri Lanka was so prolonged, there is a need to discuss the evolution of the war. Specifically, I would like to see some data (or a discussion if data is not possible) that lays out patterns of violence over time. The empirical work here only looks at a brief snapshot of the war (which was during the final phase), so it would be useful to see how this period compares to others.

[Reply]

I have added a figure (Fig 4) that demonstrates the pattern of violence (the numbers of grievous hurts and homicides) over time. Except for the outstanding increase in the number of homicides in 1988 and 1989, which may reflect the intensity of the war, there is no clear trend in the pattern of violence during the war. Although data during this period should be interpreted with caution, this graph shows that our sample period (2000–2009) is not a peculiar time period during the war. I have added this discussion to the main text.

[Comment]

• A discussion of how the suicide rate was calculated in this case would be helpful. It is very difficult to get information during wartime, even battle deaths. This implies that any suicide rates reported during wartime should be interpreted with extreme caution.

[Reply]

Although the suicide rates are gathered from several sources, all of them were originally sourced from the Registrar General’s Department. They collect data via a registrar in each registration division, each of which is further divided into administrative districts. I have added this issue to the text.

The suicide rate in Sri Lanka is known to be just about as reliable as that from many developed countries (Kearney and Miller, 1988 [21]). That said, I completely agree that the suicide rate reported during the war should be interpreted with caution. I have mentioned this in the main text. For this reason, I conducted a robustness check by dropping the districts where the census in 2002 was not conducted due to the high intensity of war, assuming that the reliability of the statistics is not high in these districts (Table 6). The estimates are still significantly negative, suggesting the robustness of the accuracy of data, albeit not perfectly.

[Comment]

How heavily is suicide stigmatized in Sri Lanka? Is there an incentive to add suicides to fatalities via the war rather than acknowledge suicide? How exactly does this process play out? I know some of these questions may not be answerable, but this has important implications for the negative relationship that is uncovered. I would like to see the author at least address this potential issue.

[Reply]

Although it is difficult to show the extent to which suicide is stigmatized in Sri Lanka, Marecek (1998) [24] mentions that the suicide of a family member exposes the presence of family problems and might diminish the marital prospects of children. In this sense, it is difficult to strictly rule out the possibility that there is an incentive to add suicide to the fatalities via the war. However, according to the Criminal Procedure Code, unnatural deaths, including suicides, are subject to inspection by a coroner before the cause of death is concluded (Fernando et al. 2003 [25]). Therefore, it is unlikely that fatalities of the war are conflated with suicide, although this possibility cannot be strictly ruled out. I have mentioned this issue in the main text.

[Comment]

• Empirically, I have two concerns. First, I am not entirely sure where the control variables come from or how they are distributed across space. Because the study is done at the local-level it is not enough to show the aggregate distribution, the author must also show local variation. Creating some maps would be great here.

[Reply]

I apologize for not mentioning this clearly. All the control variables have been sourced from Statistical Abstract, an official annual statistical report issued by the Department of Census and Statistics. Following your advice, I have created heat maps of the control variables to show variations at the district level. These maps are included in the supporting materials.

[Comment]

Second, there is notable spatial clustering in Figure 2. Has the author ruled out spatial dependence? Or, is there a need to account for diffusion in the models?

[Reply]

Thank you for this suggestion. I have added paragraphs about the analysis of spatial dependence. Specifically, I estimated a spatial autoregressive model and the spatial difference-in-difference model to control possible spillover effects from the contested area to the non-contested area (Tables 8 and 9). However, in either approach, the reduction in the suicide rate in the contested area during the war is virtually unaffected, suggesting the robustness of the findings.

---

## [Decision Letter · Decision Letter 1]

11 Aug 2020

PONE-D-20-10169R1

Revisiting suicide rate during wartime: Evidence from the Sri Lankan civil war

PLOS ONE

Dear Dr. Aida,

Thank you for submitting your manuscript to PLOS ONE. After careful consideration, we feel that it has merit but does not fully meet PLOS ONE’s publication criteria as it currently stands. Therefore, we invite you to submit a revised version of the manuscript that addresses the points raised during the review process.

Congratulations on your positive review of the revised manuscript. Please take a close look at my comments on revising some of the language in your paper. I look forward to your final touches.

We look forward to receiving your revised manuscript.

Kind regards,

Keith M. Harris, PhD

Academic Editor

PLOS ONE

Additional Editor Comments (if provided):

The manuscript is much improved and reads well. However, there are many informal expressions that don’t fit well with a scientific work. Please look at revising the following.

Line 22: “to 43..” should be “to a 43..”

Line 39: “Indeed” should be removed or changed. Also, this is not a paragraph. Paragraphs should be 3 or more sentences, consider combining some sections into full paragraphs.

Line 45: “harden ethnic minorities” this is not clear, needs clarification

Line 55: “the second important issue” this is not the best writing style, and needs clarification. Please remind readers of the issue you are writing about.

Line 63: Please state the years of the Sri Lankan war, with citation, and also clarify “the Sri Lankan suicide rate.”

Line 83 “more rigorous statistical analysis” More rigorous than what? Also, “exploiting” is not the best word here.

Line 89: It isn’t clear why this study is “more definitive.” Better to just emphasize that you have included additional factors etc.

Line 94: “the district level” that is not clear, specific that this relates to SL, and what districts are.

Line 106: The births/deaths act requires a citation and also mention that this is for SL.

Line 112: “That being said” is informal, and also not appropriate for beginning a paragraph, should be deleted.

Line 134: “Fig 1” should be spelled out in full in text. Also correct for other figure mentions.

Table 1 – this is very brief and does not require a table. Consider adding additional data or just writing out in text.

Line 178: clarify what “the contrast” refers to.

Table 3: delete “#” this can be changed to “deaths” or “total deaths” for example. It is unclear what this symbol means under “Units”

Table 5 – “R-squared” should be R2 and this does not require a lead “0”.

Line 330: “the third possible..” Again, this is not clear, specify what this refers to, and again later.

Line 387: as with the rest of the manuscript, this should be in past tense.

Line 474: “decrease in suicide rate” this needs to be clearer, annual suicide rate in SL? Specify this point.

Line 503: “It is known that..” This is informal and should be revised.

These changes will not require additional reviews.

Reviewers' comments:

Reviewer's Responses to Questions

**Comments to the Author**

1. If the authors have adequately addressed your comments raised in a previous round of review and you feel that this manuscript is now acceptable for publication, you may indicate that here to bypass the “Comments to the Author” section, enter your conflict of interest statement in the “Confidential to Editor” section, and submit your "Accept" recommendation.

Reviewer #2: All comments have been addressed

2. Is the manuscript technically sound, and do the data support the conclusions?

Reviewer #2: Yes

3. Has the statistical analysis been performed appropriately and rigorously? 

Reviewer #2: Yes

4. Have the authors made all data underlying the findings in their manuscript fully available?

Reviewer #2: Yes

5. Is the manuscript presented in an intelligible fashion and written in standard English?

Reviewer #2: Yes

6. Review Comments to the Author

Reviewer #2: I want to commend the author for taking all of my concerns seriously and putting in the hard work necessary to remedy them. I am happy with the revisions and support publication.

7. PLOS authors have the option to publish the peer review history of their article (what does this mean?). If published, this will include your full peer review and any attached files.

Reviewer #2: No

---

## [Author Response · Author response to Decision Letter 1]

23 Sep 2020

Thank you very much, once again, for your detailed comments. I have incorporated your comments as indicated below.

Line 22: “to 43..” should be “to a 43..”

[Reply] Modified

Line 39: “Indeed” should be removed or changed. Also, this is not a paragraph. Paragraphs should be 3 or more sentences, consider combining some sections into full paragraphs.

[Reply] I have changed the quotation style and combined the sentences that followed with the previous paragraph. I have also removed the word “Indeed.”

Line 45: “harden ethnic minorities” this is not clear, needs clarification

[Reply] The original expression was “harden ethnic identities” and not “harden ethnic minorities.” Hence, I have retained this expression as it is.

Line 55: “the second important issue” this is not the best writing style, and needs clarification. Please remind readers of the issue you are writing about.

[Reply] I have changed “the first important issue” and “the second important issue” to “one of the most important issues” and “another important issue,” respectively.

Line 63: Please state the years of the Sri Lankan war, with citation, and also clarify “the Sri Lankan suicide rate.”

[Reply] The years in which the war took place were stated right after the sentence, and I have added the citation for this. I have also modified “the Sri Lankan suicide rate” into “the annual suicide rate in Sri Lanka.”

Line 83 “more rigorous statistical analysis” More rigorous than what? Also, “exploiting” is not the best word here.

[Reply] I have added “than previous studies.” I have also modified “exploiting” into “estimating.”

Line 89: It isn’t clear why this study is “more definitive.” Better to just emphasize that you have included additional factors etc.

[Reply] I have added “in the sense that it provides robust evidence to possible confounding factors.” Please note that the difference-in-differences approach is different from the approaches that simply include additional factors in the regression model.

Line 94: “the district level” that is not clear, specific that this relates to SL, and what districts are.

[Reply] I have added an explanation clarifying that, in Sri Lanka, the district is the second-level administrative division after the province.

Line 106: The births/deaths act requires a citation and also mention that this is for SL.

[Reply] I have added a citation and mentioned that this is for Sri Lanka.

Line 112: “That being said” is informal, and also not appropriate for beginning a paragraph, should be deleted.

[Reply] Modified

Line 134: “Fig 1” should be spelled out in full in text. Also correct for other figure mentions.

[Reply] Modified

Table 1 – this is very brief and does not require a table. Consider adding additional data or just writing out in text.

[Reply] I have deleted Table 1 and added some text from Line 163.

Line 178: clarify what “the contrast” refers to.

[Reply] I have clarified that it is the contrast between the contested and the non-contested districts.

Table 3: delete “#” this can be changed to “deaths” or “total deaths” for example. It is unclear what this symbol means under “Units”

[Reply] Modified

Table 5 – “R-squared” should be R2 and this does not require a lead “0”.

[Reply] Modified

Line 330: “the third possible..” Again, this is not clear, specify what this refers to, and again later.

[Reply] I have modified “the possible concern” to “possible concern about the robustness of the findings.”

Line 387: as with the rest of the manuscript, this should be in past tense.

[Reply] Modified

Line 474: “decrease in suicide rate” this needs to be clearer, annual suicide rate in SL? Specify this point.

[Reply] I have specified that it is the annual suicide rate in Sri Lanka.

Line 503: “It is known that..” This is informal and should be revised.

[Reply] Modified

---

## [Editor Report · Decision Letter 2]

28 Sep 2020

Revisiting suicide rate during wartime: Evidence from the Sri Lankan civil war

PONE-D-20-10169R2

Dear Dr. Aida,

We’re pleased to inform you that your manuscript has been judged scientifically suitable for publication and will be formally accepted for publication once it meets all outstanding technical requirements.

Kind regards,

Keith M. Harris, PhD

Academic Editor

PLOS ONE

Additional Editor Comments (optional):

Thank you for your work on this manuscript and your responses to requests for minor changes.
---

## [Editor Report · Acceptance letter]

2 Oct 2020

PONE-D-20-10169R2 

Revisiting suicide rate during wartime: Evidence from the Sri Lankan civil war 

Dear Dr. Aida:

I'm pleased to inform you that your manuscript has been deemed suitable for publication in PLOS ONE. Congratulations! Your manuscript is now with our production department. 

Kind regards, 

on behalf of

Dr. Keith M. Harris 

Academic Editor

PLOS ONE